# Evaluation of a Pseudovirus Neutralization Assay for SARS-CoV-2 and Correlation with Live Virus-Based Micro Neutralization Assay

**DOI:** 10.3390/diagnostics11060994

**Published:** 2021-05-30

**Authors:** Ahmed Majdi K. Tolah, Sayed S. Sohrab, Khaled Majdi K. Tolah, Ahmed M. Hassan, Sherif A. El-Kafrawy, Esam I. Azhar

**Affiliations:** 1Special Infectious Agents Unit, King Fahd Medical Research Center, King Abdulaziz University, P.O. Box 128442, Jeddah 21362, Saudi Arabia; ssohrab@kau.edu.sa (S.S.S.); hmsahmed@kau.edu.sa (A.M.H.); saelkfrawy@kau.edu.sa (S.A.E.-K.); eazhar@kau.edu.sa (E.I.A.); 2Department of Medical Laboratory Technology, Faculty of Applied Medical Sciences, King Abdulaziz University, P.O. Box 80324, Jeddah 21589, Saudi Arabia; 3Resident Preventive Medicine, Ministry of Health (MOH), P.O. Box 109385, Al Madinah Al Munnawarah 21351, Saudi Arabia; ktolah@moh.gov.sa

**Keywords:** SARS-CoV-2, pseudotype, coronavirus, micro-neutralization assay, Saudi Arabia

## Abstract

The unusual cases of pneumonia outbreak were reported from Wuhan city in late December 2019. Serological testing provides a powerful tool for the identification of prior infection and for epidemiological studies. Pseudotype virus neutralization assays are widely used for many viruses and applications in the fields of serology. The accuracy of pseudotype neutralizing assay allows for its use in low biosafety lab and provides a safe and effective alternative to the use of wild-type viruses. In this study, we evaluated the performance of this assay compared to the standard microneutralization assay as a reference. The lentiviral pseudotype particles were generated harboring the Spike gene of SARS-CoV-2. The generated pseudotype particles assay was used to evaluate the activity of neutralizing antibodies in 300 human serum samples from a COVID-19 sero-epidemiological study. Testing of these samples resulted in 55 positive samples and 245 negative samples by pseudotype viral particles assay while microneutralization assay resulted in 64 positive and 236 negative by MN assay. Compared to the MN, the pseudotyped viral particles assay showed a sensitivity of 85.94% and a specificity of 100%. Based on the data generated from this study, the pseudotype-based neutralization assay showed a reliable performance for the detection of neutralizing antibodies against SARS-CoV-2 and can be used safely and efficiently as a diagnostic tool in a biosafety level 2 laboratory.

## 1. Introduction

The first case of unusual pneumonia was reported from Wuhan, China in late December 2019 and designated as COVID-19. The causative agent was identified similar to SARS-CoV reported in 2003 falling under the betacoronavirus group [1,2,3]. The spread of the virus caused a pandemic as declared by the WHO resulting in border closer between countries and lockdown in many parts of the world [4]. The humoral and cellular response is induced by S protein and thereby the Spike protein is being used as a significant target for vaccine and antiviral therapeutics [5,6,7,8,9,10,11,12]. The immunogenic responses to the viral diseases against S glycoprotein can be distinguished by using multiple tools including virus microneutralization (MN) assays or enzyme-linked immunosorbent assay (ELISA), and ELISA variants, such as lateral flow assay (LFA) and chemiluminescence immunoassay (CLIA) [13,14,15]. The high throughput detection of antibodies can be applied by using the ELISA with high safety and sensitivity. The major drawback of these assays is that they are not capable of detecting the neutralization ability of the positive samples [16,17,18,19]. Despite the usefulness of SARS-CoV-2 MN assays and their ability to detect neutralizing antibodies and continue to be the gold standard method, the MN assay needs a biosafety-level-3 facility and well-trained personnel with specialized skills [20]. 

Currently, there are different types of pseudotyping assays against SARS-CoV-2 that have now been developed using HIV-based lentiviral [21], MLV-based retroviral [22,23], and vesicular stomatitis virus (VSV) [24]. The correlation of measurements performed using these assays was significantly similar with microneutralization assays using live SARS-CoV-2 [23,25]. The currently developed system such as lentiviral pseudotype has been efficiently adopted and used against emerging and re-emerging viruses and has been proven as a safe and versatile tool to study the efficacy of antiviral therapies and vaccines [26,27,28]. The use of SARS-Co-2 Spike protein based pseudotypes virus particles provides a useful tool for the detection of antibodies as well as for the evaluation of neutralization capacity of antibodies in serum samples of COVID-19 patients [29,30].

In this study, we have evaluated the performance of pseudotype lentiviral particles using the Spike gene of SARS-CoV-2 and to validate the efficacy of pseudotype particles assay in detecting neutralizing antibodies against SARS-CoV-2 in serum samples collected in a sero-epidemiological study from Jeddah, Saudi Arabia. The results of the assay were compared to microneutralization assay.

## 2. Materials and Methods

### 2.1. Culture of Cell Lines

Vero E6 (ATCC CRL-1586) and HEK 293 T/17 cells (ATCC–CRL 1573) were obtained from the American Type Culture Collection (ATCC). HEK 293T cells with human ACE2 expression (NR-52511) were kindly provided from the BEI resources, NIAD, and NIH. All the cell lines were grown in standard DMEM with 10% FBS at 37 °C and 5% CO_2_.

### 2.2. Virus Isolate

A SARS-CoV-2 clinical human isolate (GenBank accession number: MT630432) was used in all experiments at the Special Infectious Agents Unit (SIAU), King Fahd Medical Research Center, King Abdulaziz University, Jeddah, Saudi Arabia.

### 2.3. Serum Samples

This study was approved by the Research Ethics Committee (REC), Unit of Biomedical Ethics, Faculty of Medicine, King Abdulaziz University (KAU) (Reference No 35–21). Written consent forms were obtained from participants. Serum samples used in this study were received in SIAU as a part of a seroprevalence study of SARS-CoV-2 (*n* = 300). Antibodies to SARS-CoV-2 were originally tested using micro-neutralization (MN) assay conducted in the BSL-3 facility of SIAU. These sera were further examined with SARS-CoV-2 pseudotypes to evaluate the performance of the assay against the live virus assay in assessing the neutralization titer of the serum samples.

### 2.4. Microneutralization (MN) Assay

The collected serum samples were verified for neutralizing antibodies (nAbs) using published protocol [31]. Briefly, sera were heat inactivated and diluted to an initial dilution of 1:20 followed by a serial dilution of 1:2 in DMEM supplemented with 2% fetal bovine serum (FBS) and antibiotics. The diluted samples were co-incubated for 1 h at 37 °C with SARS-CoV-2 isolate (100 TCID_50_) in a 1:1 ratio (*v/v*). The mixture was then added to a monolayer of Vero E6 cells in a 96-well plate and incubated at 37 °C in 5% CO_2_ for three days and observed daily for any cytopathic effect (CPE). All the experiments were performed quadruplicates and samples with CPE in all wells were considered as positive, and titer was defined as the first dilution showing CPE in all wells.

### 2.5. Generation of Pseudotyped Lentiviral Particles

The plasmids used were supplied from BEI Resources [32]. The HEK 293T/17 cells (2 × 10^5^ cells/mL) were seeded into six-well plates in DMEM with 10% FBS till they have a 50–70% confluency. At 16–24 h, the cells were transfected with the plasmid DNA for lentiviral production. The transfection was performed using Lipofectamine TM 2000 following the manufacturer’s instructions. The complex mixture was made by adding the lentiviral backbone DNA (1 µg-Luciferase-IRES-ZsGreen) and 0.22 µg of each plasmid DNA (HDM-Hgpm2, pRC-CMV-Rev1b, HDM-tat1b) while 0.34 µg plasmid DNA of SARS-CoV-2 Spike protein was used as viral entry protein. Lentiviral virus was used as positive control and cell only wells were used as negative control. The cells were replenished with fresh DMEM containing 10% FBS after 18–24 h post transfection and further incubated for 48 h at 37 °C and 5% CO_2_. After 48 h, the supernatants were collected, filtered, and finally stored at −80 °C for further use.

### 2.6. SARS-CoV-2 Pseudotypes Titration

In a 96-well white opaque culture plate, 100 ul of supernatant contain SARS-CoV-2 pseudotype virus particles were added, positive control lentiviral virus, and negative control cell only, followed by a 1:2 serial dilutions. HEK293-ACE2 cells from the 75 cm^2^ tissue culture flask were harvested, and the cells were counted to 2 × 10^5^ cells/mL. Finally, 50 μL of cells on 50 μL of diluted pseudotype virus particles were added, and the plate was centrifuged for 1 min at 500× *g* and incubated for 48 h at 37 °C, 5% CO_2_. After incubation, a Bright-Glo assay system is used (E2610, Promega, Madison, WI, USA) for luminescence generation. Then, luminescence is measured in Relative Luminescence Units (RLUs) using a plate reader (Synergy 2, BioTeck). RLUs are then plotted against virus dilution. In calculating the virus titer, the wells that showed >1000-fold signal above a virus-only background were selected, and the titration curve was plotted which showed a linear relationship between virus titer and RLU (Figure 1).

### 2.7. Pseudotype-Based Neutralization Assays

The pseudotype-based neutralization assays were performed following the published protocol [33] with minor modifications. In brief, the serum samples were initially diluted to a 1:20 dilution followed by a 1:1 serial dilution. Diluted serum samples were mixed with SARS-CoV-2 pseudotypes in a 1:1 *v/v* ratio and then incubated for 1 h at 37 °C in a 5% CO_2_ atmosphere. After 1 h, the transfected cell suspensions were mixed into each well of cell culture plate and further incubated for 48–72 h at 37 °C and the neutralizing antibodies were characterized based on luciferase activity reading.

### 2.8. Calculation of SARS-CoV-2 Pseudotype Titers

The neutralization titers of samples were calculated by first normalizing the neutralization titer for each dilution relative to the positive control; then, the IC_50_ was calculated through a nonlinear regression model of log10 inhibitor dilution vs. normalized response with a variable slope using GraphPad Prism version 9.0 package (GraphPad Software, San Diego, CA, USA). Titers were then expressed as the range of sample dilution in which the IC_50_ value lay and the dilution closest to the IC_50_ value was chosen as the antibody titer.

### 2.9. Sensitivity and Specificity of the Assay

The sensitivity of the assay compared to the MNT assay as the reference assay was calculated according to the following equation:Sensitivity = (number of positive samples/(number of positive samples + number of false negative samples)) × 100,
while the specificity of the assay compared to the MNT assay as the reference assay was calculated according to the following equation:Specificity = (number of negative samples/(number of negative samples + number of false positive samples)) × 100

## 3. Results

### 3.1. Sero-Status of Sample

The samples were tested using SARS-CoV-2 MN assay as the reference standard with a MN titer of >1:20 used as the cut-off for positive samples. The MN assay showed that 64 samples were positive while 236 samples were negative.

### 3.2. Titration of the Generated SARS-2-S-Luciferase Pseudoviruses

To measure the titers of lentiviral pseudotyped particles, we produced viral particles with SARS-CoV-2 spike gene, and quantification was performed using a luminescence signal of the expressed luciferase. Based on the RLU values observed at each serial dilution, virus dilution vs. RLU readout was plotted to select the needed amount of virus for further experiments (Figure 1), and the signal for the needed amount of virus was found to be the quantity producing a luminescence of 10^4^ RLUs per mL in 96-well plate infections.

### 3.3. Neutralization Assays with Spike-Pseudotyped Lentiviral Particles

The assay was conducted by using the confirmed positive and negative samples with MN assay. A serial dilution of serum samples was made in a 96-well plate and incubated with pseudotyped lentiviral particles for sixty minutes to achieve 2 × 10^4^ RLUs. Finally, the mixture was added to a pre-seeded plate of 293T-ACE2 cells, and the luciferase activity was measured 48 h post-infection. Dilution-dependent reduction in luciferase activity was observed with a tested seropositive sample, in contrast to the seronegative sera that did not show significant changes in the luciferase activity with dilution. Neutralizing Ab titer was computed as IC_50_ using a 4PL logistic curve fitting in Prism 9 (GraphPad, San Diego, CA, USA).

### 3.4. Correlation between Live Virus MN and Pseudotype-Based Neutralization Assays

Out of 300 serum samples analyzed, we found that only 55 positive samples and 245 negative samples by pseudotyped viral particles assay, while 64 were positive and 236 were negative by MN assay (Table 1) (Figure 2). Therefore, only nine serum samples were found to be false negative when compared with MN results as shown in (Table 2). Compared to the microneutralization assay, the pseudotyped viral particles assay showed a sensitivity of 85.94% and a specificity of 100%. A simple linear regression was made to predict the log2 PNT based on the log2 MNT data (Figure 3).

Results show the concordance of antibody titers between MNT and PNT in 16 samples as follows: 4 samples had a titer of 1:80, 7 samples with a titer of 1:160, 3 with a titer of 1:320, and 1 each with a titer of 1:640 and 1:1280. PNT was constantly showing at least one dilution higher than MNT except for two samples, where it showed one dilution lower than MNT.

## 4. Discussion

The pseudotyped viral particles assays offer the advantage of evaluating the titer of the antibodies without the need for the laboratory intensive and the biohazard safety measures needed for the microneutralization assays, especially for highly contagious pathogens like SARS-CoV-2. The assay also offers a more quantitative measure based on the RLU reading rather than relying on the visual evaluation of the observer under the microscope.

The emergence of COVID-19 led to a global pandemic, exposing a serious threat to human health. SARS-CoV-2 infection results in the induction and elicitation of specific antibodies that bind to SARS-CoV-2 in infected patients. Recent publications have shown that effective protection against SARS-CoV-2 infection (or re-infection) is achieved through several factors including cellular immunity and the level of neutralizing antibodies in the sera of cases [33]. Currently, the diagnosis of SARS-CoV-2 infection is being performed by many standard assays [16,17,18,19]. However, the demand for serological assay is always valuable and very high. This demand is needed for sero-epidemiological studies and the screening of a large number of samples including symptomatic, non-symptomatic, vaccinated individuals as well as recovered patients. Additionally, serological assays are also a very valuable, low-cost tool as well as being available in laboratories with limited facilities. Detection of antibodies using ELISA provides a qualitative measure of antibodies to SARS-CoV-2 and does not show the neutralizing ability of the detected antibodies. The standard method for the detection of neutralizing antibodies is the MNT, which requires the use of a live SARS-CoV-2 virus and the availability of a BSL-3 laboratory. These limitations of the ELISA and MNT have encouraged us to evaluate the pseudotyped viral particles assay using the spike protein of SARS-CoV-2 for the detection of neutralizing antibodies in serum samples. The developed assay was evaluated using human serum samples and further validated by comparison with the MN assay.

The assay was setup as previously described by Crawford et al. [26], and the neutralization titer was estimated by serially diluting the samples and evaluating the range of dilution where the IC_50_ falls between. The signal generated by the amount of lentivirus generated was found to be 10^4^ RLUs per mL in 96-well plates, which is in accordance with the data reported by Crawford et al. [26]. The assay showed a robust performance with an inter-run assay variability of <10% and an intra-run assay variability of <3%.

Our data showed an 85.9% sensitivity and 100% specificity of the assay when compared to the standard MN assay. The reduction in assay sensitivity was because of nine samples that showed false negative results by PNT and showed a low titer in MN assay (1:40) [33]. These data are in accordance with the data reported by Hyseni et al. [33], where they compared the performance of PNT compared to MN assay by testing 65 human serum samples, 28 samples were positive by MN, while 24 samples were positive by PNT. On other hand, 37 negative samples by MN and 41 samples were negative by PNT. Comparable to our study, they reported a sensitivity of 85.7% and the specificity of 100% [33].

## 5. Conclusions

Based on the data generated from this study, the assay can be used with confidence in the sensitivity and specificity for the evaluation of the titer of neutralizing antibodies against the spike protein of SARS-CoV-2 in infected and vaccinated cases. This assay has the advantage of being used in a BSL-2 facility and does not require a high-level biocontainment facility because the pseudotype is devoid of virulent viral components.

## Figures and Tables

**Figure 1 diagnostics-11-00994-f001:**
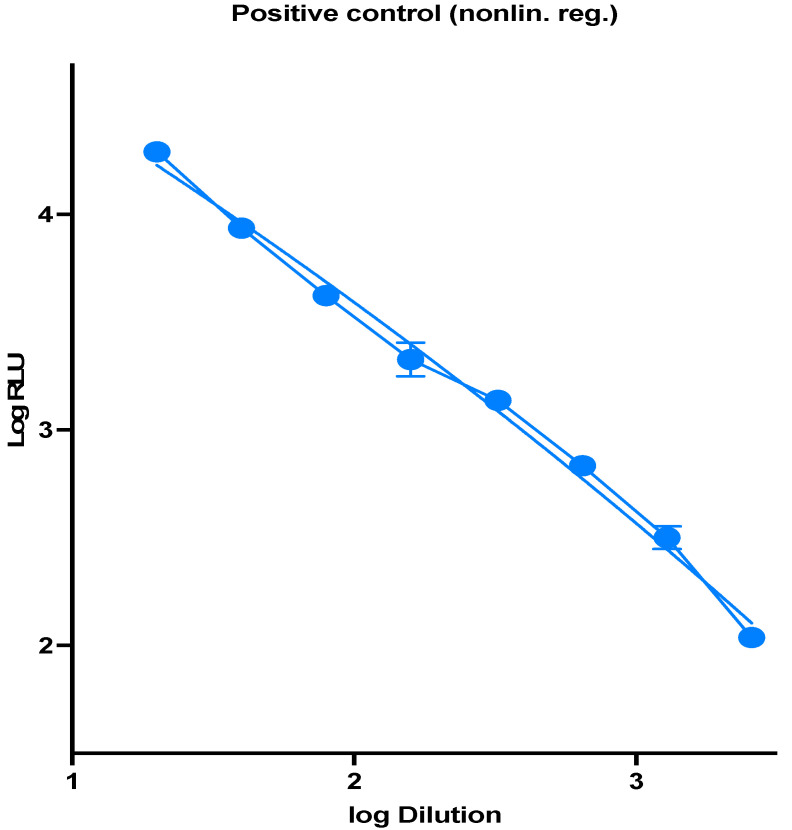
RLU values observed at each serial dilution, virus dilution vs. RLU readout was plotted to select the needed amount of virus.

**Figure 2 diagnostics-11-00994-f002:**
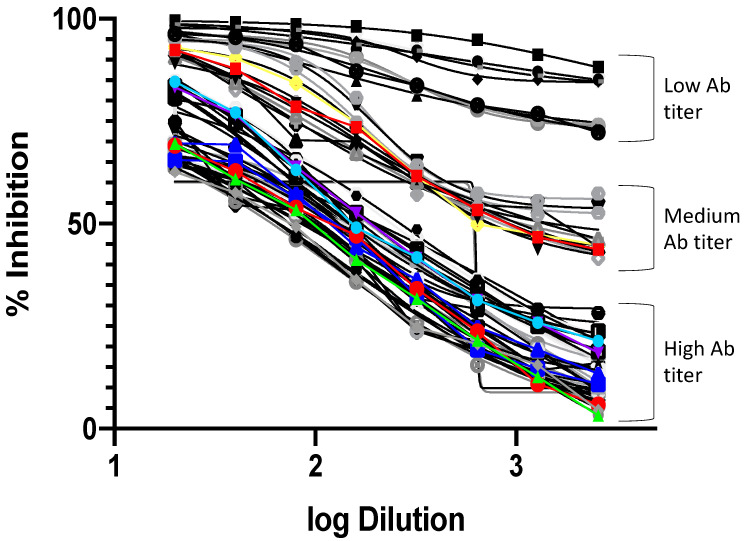
Dilution curves for the positive samples included in the study. The curves are plots of the log sample dilution vs. the percent normalized inhibition for each dilution compared to the positive control. The curves show samples with high, medium, and low titers.

**Figure 3 diagnostics-11-00994-f003:**
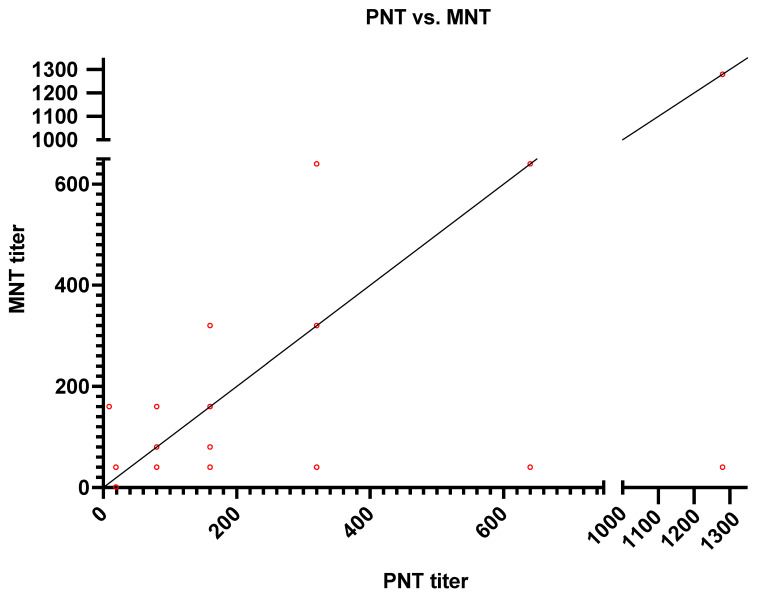
Correlation of SARS-CoV-2 Ab titers as detected by PNT (*x*-axis) vs. MNT (*y*-axis).

**Table 1 diagnostics-11-00994-t001:** Total number of serum samples by MN and PNT.

	MNT	PNT
Negative	236	245
Positive	64	55
Total	300	300

**Table 2 diagnostics-11-00994-t002:** Sensitivity and specificity of the pseudotyped viral particles assay.

	MNT	
PNT	Positive	Negative
Positive	55	0
Negative	9	236
Sensitivity	85.94%	
Specificity	100%	

## Data Availability

All data that support the findings of this study are included in this manuscript.

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
