# Peer review of "Evaluation of a Pseudovirus Neutralization Assay for SARS-CoV-2 and Correlation with Live Virus-Based Micro Neutralization Assay"

_diagnostics, 2021, doi:10.3390/diagnostics11060994_

Round 1
Reviewer 1 Report
In the current study, Tolah et al discussed the usefulness of pseudootyped SARS COV2 particle as a model for detection of neutralizing antibodies. The study is interesting but extensive revisions are needed regarded the description of methods, the presentation of figures, the results and the discussion parts. Also negative samples (maybe collected prior to the pandemic) should be included in the evaluation study.
Reviewer 2 Report
The article by Tolah et al describe a set of data that directly compare the pseudo virus neutralization titer (PNT) assay to the micro-neutralization MN assay for 300 serum samples. The experimental design are well done and results are generally presented. The conclusions seem to support the observations from other groups as well that PNT is a robust method for testing neutralization of SARS-CoV2. I would recommend the authors to consider re-formatting the following:
- Abstract: the first 4 sentences are not connected to each other, there is no flow of information. The results and assay description are poorly included in abstract. Re-format the abstract with emphasis on PNT and MN assay and describe the results and conclusion
- Introduction: several sentences are repetitive (the origin and classification of SARS-CoV2) this could be written in a simpler context and bringing the readers attention to the real message of the paper PNT and MN assays. Also please describe in detail the principle behind the PNT assay and MN assay. Introduce the PNT assay better in lines 59-69.
- The methods section: what was the dynamic range in the luminometer used? please elaborate the sections pertaining to MN assay and PN-titer measurement, somewhat unclear to me.
- Results, generally acceptable, but could have been more elaborate. especially when presenting figures:
- the numbers on x and y-axis are too small to read Fig-1 and Fig-3
- the figure legends should be more descriptive to help the reader to understand
- What is the basis of selecting >1:20 as threshold on MN assay, I assume this is because of the initial dilution used in the assay
- the authors say that RLUs >1000 of blank control was used as a positive result, then in line 161 mention 104 RLUs please clarify what this 104 means? Also in lies 157-162 “needed amount of virus” is not informative. Please explain clearly what amount of virus was used.
- Also through out the paper check sentences for grammar and spelling. Some of which I have highlighted in attached file.

Round 2
Reviewer 1 Report
The resubmitted manuscript has improved and the comments have been addressed accordingly.